# Effects of *Bacillus cereus* on Survival, Fecundity, and Host Adaptability of Pine Wood Nematode

Yuchao Yuan †, Zhengmei Yan †, Yangxue Chen, Jianren Ye and Jiajin Tan *

College of Forestry, Nanjing Forestry University, Collaborative Innovation Center for Sustainable Forestry in Southern China, Nanjing 210037, China
* Correspondence: tanjiajin@tom.com
† These authors contributed equally to this work and share first authorship.

**Abstract:** To clarify the role of bacteria in PWD, three PWNs with different virulence (strongly virulent strain AMA3, normally virulent strain AA3, and weakly virulent strain YW4) were selected as research objects, and three strains of *Bacillus cereus* (nematode-associated bacteria GD1, *Pinus massoniana* endophytic bacteria GD2, and *P. elliottii* endophytic bacteria NJSZ-13) at different concentrations were used to determine their effects on the survival and fecundity of the nematodes. The results showed that strains GD1 and GD2 could significantly improve the survival and fecundity of PWNs at three different concentrations, while NJSZ-13 showed the opposite effects. The inoculation experiments showed that the disease index of *P. massoniana* under different treatments was as follows: AMA3 < a mixture of AMA3 and GD1 < a mixture of AMA3 and GD2. Similar results were shown in the *Larix kaempferi* inoculation experiment. Further, using RNA-sequencing analysis, we found that the up-regulated genes in PWN were *sHsp 21*, *Hsp 70*, and *Hsp 72* after being treated by strains GD1 and GD2. The longevity regulatory pathways, MAPK signaling pathways, glutathione metabolic pathways, and cytochrome *P450* metabolic pathways related to these genes are clearly enriched. These results show that the bacteria can improve the host adaptability of PWN, and endophytic bacteria of pine trees may be more effective in improving the host adaptability of nematodes than the associated bacteria of nematodes.

**Keywords:** *Bursaphelenchus xylophilus*; PWN-carried bacterium; endophytes; survival; fecundity; host adaptability





## 1. Introduction

The pine wood nematode (PWN) *Bursaphelenchus xylophilus* is one of the most dangerous and destructive pests in global forest ecosystems [1], and has the characteristics of rapid onset, rapid spread, and difficult control, causing serious damage and huge economic loss to the world's forest resources [2]. It is mainly distributed in the United States, Canada, and Mexico in North America [3,4]; Japan, China, and South Korea in Asia [5]; and Portugal and Spain in Europe [6,7]. At present, 60 million hm$^2$ of pine forests are facing the threat of the pine wilt disease (PWD) in China [8]. PWNs are known to endanger species of the *Pinus genus* including: *P. massoniana*, *P. tabulaeformis*, *P. densiflora*, *P. armandii*, *P. thunbergii*, *P. sylvestris* var. *mongolica* Litv., and other pine species present in China [9,10]. In December 2018, PWD was reported on non-*Pinus* host *Larix gmelinii* in China [11]. *Larix gmelinii* belongs to the genus *Larix*, which is the main tree species of alpine coniferous forest in Northeast China, Inner Mongolia, North China, and Southwest China. Infection of a large area of pine forest still causes irreparable damage. The high pathogenicity of PWNs in non-*Pinus* species enhances the risk of PWN transmission to other parts of the globe. The pathogenic mechanism of PWNs has always been controversial, and the role of bacteria in this complex pathogenic mechanism continues to be a focus of study.

Oku et al. [12] first proposed that PWD was caused by a wilt toxin produced by the bacterial genus *Pseudomonas* carried by PWNs. Kawazu et al. [13] isolated three strains

of the bacterial genus *Bacillus* (*B. subtilis*, *B. cereus*, and *B. megaterium*) from the body of PWNs, and pointed out for the first time that phenylacetic acid was the pathogenic toxin of PWD. Zhao et al. [14] suggested that PWNs and their toxin-producing bacteria compound led to PWD. Xie and Zhao [15] found that there was a certain correlation between bacteria and PWN population dynamics in different stages of PWD. In addition, Tian et al. [16] conducted macrogenomic analysis and discovered that the bacteria on the body of PWNs are highly diverse. Interestingly, in different countries, microbial communities found on the body of PWNs or infected pine trees are also markedly dissimilar and diverse [17]. Further analysis showed that PWNs can produce cellulase (β-1,4-endoglucanase) and pectinase to invade plant tissues, which may be caused by horizontal gene transfer from bacteria and fungi to nematodes [18]. This unexpected coevolution process provides a possible theoretical basis for the study of the interaction between PWNs and bacteria. So far, studies have shown that PWN-related bacteria can significantly improve their adaptability, such as terpene adaptability, ethanol adaptability, and low-temperature tolerance [19–21]. It is worth mentioning that conditional pathogenic bacteria, plant beneficial bacteria, or plant pathogenic bacteria may be associated with PWD, so it is important to explore the exact role of bacteria in this interaction.

Although the interaction between PWNs and specific bacteria (PWN-associated bacteria and pine endophytic bacteria) has been well documented, less is known about the relationship between PWNs with different virulence and bacteria. In addition, there are few reports on the differential gene expression of PWN treated with different bacteria. Although the pathogenicity of PWNs to different pine hosts and cedar hosts has been reported, the pathogenicity of PWNs and bacteria to larch has not been reported. Hence, in this study, we aimed to: (1) Investigate the correlation between PWN-carried bacteria, pine endophytic bacteria, bacteria with nematicidal activity, and PWNs; (2) Explore the effect of bacteria on the adaptability of PWN in two different hosts; (3) Analyze the effect of bacteria on the transcriptome of PWNs. It is of great significance to further identify the possible influencing factors of PWN virulence and to explore the interaction between PWNs and bacteria.

## 2. Materials and Methods

### 2.1. PWN Strains, Bacterial Strains, and Hosts

Three populations of *B. xylophilus* were used, they had been maintained as different virulence isolates in our laboratory: high virulence AMA3 (Maanshan, China), intermediate virulence AA3 (Anqing, China), and low virulence YW4 (Dehong, China). These populations have been used in various studies as different virulence isolates [22–25]. In addition, three strains of bacteria were used: the associated bacteria GD1 isolated from the body of PWNs, the endophytic bacteria GD2 isolated from healthy *P. massoniana*, and the endophytic bacteria NJSZ-13 with nematicidal activity isolated from *P. elliottii* [26,27]. All three isolates are *Bacillus cereus*. All PWNs and bacterial strains were stored in the Forest Protection Laboratory, Nanjing Forestry University (Nanjing, China). In addition, the strain NJSZ-13 was deposited as no. M2016660 in the China Center for Type Culture Collection (Wuhan, China). The tested pine seedlings were two-year-old *P. massoniana* (Hechi, China) and two-year-old *L. kaempferi* (Dalian, China).

### 2.2. Pretreatment of Nematodes and Bacteria

Three nematode populations with different virulence were propagated on *Botrytis cinerea* cultured on potato dextrose agar (PDA) plates for 4–5 days at 25 °C. The nematodes were isolated overnight using the Baermann funnel technique [28]. The nematode suspension was put into a sterile 15 mL centrifuge tube, with 3000 r/min centrifugation for 3 min. Then the supernatant was discarded, treated with 3% hydrogen peroxide for 15 min, centrifuged, next treated with 0.02% streptomycin sulfate for 30 min, and then washed and centrifuged with aseptic sterilized deionized water 3 times. Single colonies GD1, GD2, and NJSZ-13 were obtained from the nutrient agar medium, put into 150 mL flasks with 50 mL

of nutrient broth medium, and shaken at 200 r/min at 27 °C for 12 h. Then, the colonies were put it into aseptic 50 mL centrifuge tubes and centrifuged at 6500 r/min for 3 min. The supernatant was discarded and washed with aseptic deionized water 3 times. The bacterial concentration was counted by the combination of the dilution plating procedure and turbidimetry [29].

### 2.3. Viability of Nematodes with Contrasting Virulence under Different Bacterial Treatments

Under aseptic conditions, 2000 PWNs were mixed with different concentrations of $5 \times 10^6$, $5 \times 10^5$, and $5 \times 10^4$ CFU/mL of the strains GD1, GD2, and NJSZ-13 in aqueous suspension (2 mL), respectively. The treatment groups were named AMA3 (control), AMA3 + GD1, AMA3 + GD2, and AMA3 + NJSZ-13. AA3 and YW4, which used the same processing method, were named as AA3 (control), AA3 + GD1, AA3 + GD2, AA3 + NJSZ-13, and YW4 (control), YW4 + GD1, YW4 + GD2, and YW4 + NJSZ-13, respectively. All of the 12 treatment groups were placed in a gradient incubator at 25 °C. A total of 0.05 mL of the mixture was observed every two days, and the survival rate of nematodes was determined under a light microscope (Leica DM500, Wetzlar, Germany). Use the following formula to calculate the nematode survival rate of each treatment group. Nematode survival rate (%) = $\frac{\text{Number of live nematodes}}{\text{Total number of nematodes}} \times 100$. The live or dead nematodes were determined by pricking with a needle. With aseptic sterilized deionized water as the control, each treatment group was repeated five times.

### 2.4. Fecundity of Different Virulent Nematodes Treated with Different Concentrations of Bacteria

Under aseptic condition, 5000 treated PWNs of AMA3, AA3, and YW4 were mixed with $5 \times 10^6$, $5 \times 10^5$, and $5 \times 10^4$ CFU/mL strains of GD1, GD2, and NJSZ-13 in aqueous suspension (2 mL), respectively. The mixture of 0.2 mL bacteria and nematodes was absorbed into the sterilized cotton ball (about 5 mm in diameter) in the center of the PDA plate full of *Botrytis cinerea* using the cotton ball method [30]. After being cultured at 25 °C for five days, the nematodes were isolated and counted. With aseptic sterilized deionized water as the control, each treatment was repeated five times.

### 2.5. Inoculation of P. massoniana and L. kaempferi with Mixed Inoculation of Bacteria and Nematodes

The pretreatment methods were same as in Section 2.2, in which the treatment of AMA3 and strains of GD1 and GD2 occurred. The two-year-old *P. massoniana* and *L. kaempferi* were inoculated using the artificial stem-cutting inoculation method [5]; six groups which include five treatments groups and one control group were set up. Ten *P. massoniana* were inoculated in each group: (1) AMA3 + GD1: inoculated with a mixture of 5000 PWNs/mL and $5 \times 10^7$ CFU/mL of strain GD1 in a total of 2 mL; (2) AMA3 + GD2: inoculated with a mixture of 5000 PWNs/mL and $5 \times 10^7$ CFU/mL of strain GD2 in a total of 2 mL; (3) AMA3: inoculated with 2 mL of 5000 PWNs/mL; (4) GD1: inoculated with $5 \times 10^7$ CFU/mL of strain GD1 in a total of 2 mL; (5) GD2: inoculated with $5 \times 10^7$ CFU/mL of strain GD2 in a total of 2 mL (6) The total dose of sterile deionized water inoculated with 2 mL was used as the control group. The treatment group and control group of *L. kaempferi* was the same as that of *P. massoniana*, except the number of *L. kaempferi* inoculated in each treatment group was 5, and *L. kaempferi* was inoculated twice. When inoculating, first, a wound was cut obliquely with a sterilized scalpel 6–8 cm above the rhizome of the host; a sterilized cotton ball was then used to plug the wound and fix the wound with a funnel-shaped parafilm sealing membrane. The mixture of bacteria and nematodes, which had been diluted and counted in advance, was dropped onto the cotton ball, moisturized with aseptic water regularly, and cultured in a greenhouse under a daytime temperature of 28 °C during the experimental period.

The symptoms of the inoculated hosts were observed every five days, and the disease incidence and disease index (DI) were calculated. The disease index reflected the difference in wilting degree of indirect species among different treatments, and the disease incidence

reflected the difference in wilting number among repeats of each group. The severity of infection in *P. massoniana* and *L. kaempferi* was divided into five levels (Table 1). The disease grading standard and DI were calculated according to Yu et al. [31].

$$\text{Disease incidence } (\%) = \frac{\sum \text{number of infected plants with symptoms}}{\text{Total number of plants}} \times 100 \qquad (1)$$

$$\text{Disease index} = \frac{\sum \text{number of diseased plants } \times \text{ disease grade}}{\text{Total number of plants } \times \text{ highest disease grade}} \times 100 \qquad (2)$$

**Table 1.** Disease grade of pine wilt disease in *P. massoniana* and *L. kaempferi*.

| Disease Grade | Grading Standard | Representative Value |
|:---:|:---:|:---:|
| I | Healthy, normal growth of plants | 0 |
| II | A few needles turned yellow | 1 |
| III | Half of the needles turned yellow and the branches were bent | 2 |
| IV | Most of the needles turned yellow, tree became wilted | 3 |
| V | All the needles turned yellow, tree wilted | 4 |

*2.6. RNA Isolation, cDNA Synthesis, and RNA Sequence*

Under aseptic condition, 5000 treated PWNs of AMA3 were mixed with $5 \times 10^6$ CFU/mL of strains GD1 and GD2 in aqueous suspension (2 mL), respectively. Three groups were named CK(AMA3), T1(AMA3 + GD1), T2(AMA3 + GD2), respectively. Total RNA was extracted using a Trizol reagent kit (Invitrogen, Carlsbad, CA, USA) according to the manufacturer's protocol. RNA quality was assessed using an Agilent 2100 Bioanalyzer (Agilent Technologies, Palo Alto, CA, USA) and checked using RNase free agarose gel electrophoresis. Then, the enriched mRNA was fragmented into short fragments using fragmentation buffer and reverse transcribed into cDNA with random primers. Second-strand cDNA were synthesized using DNA polymerase I, RNase H, dNTP, and buffer. Then, the cDNA fragments were purified with a QiaQuick PCR extraction kit (Qiagen, Venlo, The Netherlands), end repaired, had poly(A) added, and ligated to Illumina sequencing adapters. The ligation products were size-selected by agarose gel electrophoresis, PCR amplified, and sequenced using the Illumina HiSeq2500 by Gene Denovo Biotechnology Co. (Guangzhou, China).

*2.7. Quantitative Real-Time PCR*

Contaminating genomic DNA was removed using treating total RNA with gDNA wiper (Vazyme, Nanjing, China). The cDNA was synthesized using the HiScript II qRT SuperMix II (Vazyme, Nanjing, China). The q-PCR reaction mixture (total 20 µL) contained 10 µL of ChamQ SYBR qPCR Master Mix (Vazyme, Nanjing, China), 0.4 µL of each forward and reverse primer, 7.2 µL of ddH$_2$O, and 2 µL of template cDNA. The q-PCR conditions were as follows: pre-denaturation at 95 °C for 30 s, followed by 40 cycles of denaturation at 95 °C for 10 s, and annealing at 60 °C for 34 s. The primer sequences were listed in Table 2. Actin (GenBank EU100952) was chosen as a reference for q-PCR. All samples were run in triplicate, and gene expression levels were quantified using the $2^{-\Delta\Delta Ct}$ method. Three biological replicates and three technical replicates were performed for each treatment.

**Table 2.** The RT–qPCR primers for 8 differentially expressed genes.

| ID | Forward Primer (5′–3′) | Reverse Primer (5′–3′) |
|---|---|---|
| BXY_1563600 | AAGGGCCGTCTCTCACAAAG | TCTCTGCCGTCTGGTTGTTC |
| BXY_0640100 | TCAATGGGTGGAGAGCAACC | CAGTAGGTCCACTGGCTTGG |
| BXY_1045400 | TGGAGGCAATTCAGGCTCAG | ACTCGGAGCCCAACGAATTT |
| BXY_0165400 | ACCGACACATCAGGATTCCG | GGGCTTCACTTGAATGGGGA |
| BXY_0727300 | CCCACAATGTCGCCAATCCT | CCACATCAGCGGGAAGGAAA |
| BXY_0496700 | GCCTTTCGCTGGAAGACCC | AACCCTCGTCGCACTGTCG |
| BXY_1091200 | CCGTGCCTGCTCATCATTCT | ATCCCGACCTGCTTACAACG |
| EU100952 | GCAACACGGAGTTCGTTGTA | GTATCGTCACCAACTGGGAT |

*2.8. Sequencing Data Analysis*

To get high quality clean reads, reads were further filtered using fastp (version 0.18.0). The parameters were as follows: (1) Remove reads containing adapters; (2) Remove reads containing more than 10% of unknown nucleotides (N); (3) Remove low quality reads containing more than 50% of low quality (Q-value $\leq$ 20) bases.

RNAs differential expression analysis was performed using DESeq2 software. The genes with the parameter of a false discovery rate (FDR) below 0.05 and absolute fold change $\geq$ 2 were considered to be differentially expressed genes.

RNA sequencing data have been submitted to the National Center for Biotechnology Information (NCBI) under BioProject PRJNA760241.

*2.9. Statistical Analysis*

All data were expressed as means $\pm$ standard error of means (SEM). All parameters were calculated using Microsoft Excel. The analysis of the variance plus Tukey's test was performed using Prism 6 software (Graphpad, San Diego, CA, USA). The differences were considered statistically significant at $p < 0.05$.

## 3. Results

*3.1. Survival of Nematodes with Different Virulence under Different Bacterial Treatments*

When different concentrations of GD1, GD2, and NJSZ-13 were mixed with high virulent AMA3, the survival rate of nematodes decreased gradually, similar to the control group. The survival rate of nematodes under different concentrations of the AMA3 + GD1 treatment group was higher than that of the AMA3 + GD2 treatment group, while the result of the control group was lower than that of the AMA3 + GD1 and AMA3 + GD2 treatment groups, and higher than that of the AMA3 + NJSZ-13 treatment group (Figure 1A–C). Under the treatment of $5 \times 10^4$ CFU/mL strains, the result of the control group, AMA3 + GD1, and AMA3 + GD2 treatment groups decreased most rapidly between day 6 and day 10 (Figure 1C). Under the treatment of $5 \times 10^5$ CFU/mL and $5 \times 10^6$ CFU/mL strain concentrations, the result of the AMA3 + GD1 and AMA3 + GD2 treatment groups decreased most rapidly between day 10 and day 14 (Figure 1A,B). Under the treatment of $5 \times 10^5$ and $5 \times 10^6$ CFU/mL strains, the survival rate of nematodes in the control group decreased most rapidly between day 6 and day 10 (Figure 1A,B).

When different concentrations of GD1, GD2, and NJSZ-13 were mixed with intermediate virulent AA3, the nematode survival decreased gradually, just like the control group AA3. The AA3 survival under different concentrations of the AA3 + GD1 treatment group was higher than that of the AA3 + GD2 treatment group, while the AA3 survival of the control group was lower than that of the AA3 + GD1 and AA3 + GD2 treatment groups, and higher than that of the AA3 + NJSZ-13 treatment group (Figure 1D–F). Under the treatment of $5 \times 10^4$, $5 \times 10^5$, and $5 \times 10^6$ CFU/mL strains, the survival of the control group AA3, and the treatment groups AA3 + GD1 and AA3 + GD2, decreased most rapidly between day 10 and day 14 (Figure 1D–F). When different concentrations of GD1, GD2, and NJSZ-13 were mixed with low virulent YW4, the nematode survival decreased gradually, similar to the control group YW4. The YW4 survival under different concentrations of the YW4 + GD1 treatment group was higher than that of the YW4 + GD2 treatment group,

while the YW4 survival of the control group was lower than that of the YW4 + GD1 and YW4 + GD2 treatment groups, and higher than that of the YW4 + NJSZ-13 treatment group (Figure 1G–I). Under the treatment of $5 \times 10^4$ CFU/mL strains, the survival of the control group YW4, YW4 + GD1, and YW4 + GD2 treatment groups decreased most rapidly between day 10 and day 14 (Figure 1I). Under the treatment of $5 \times 10^5$ CFU/mL strains, the survival of the control group YW4 and YW4 + GD2 treatment groups decreased most rapidly between day 10 and day 14 (Figure 1H). Under the treatment of $5 \times 10^5$ and $5 \times 10^6$ CFU/mL strains, the YW4 + GD1 survival decreased most rapidly between day 14 and day 18 (Figure 1G,H). GD1 and GD2 could significantly improve the survival of PWNs at three concentrations, while NJSZ-13 showed the opposite result. The three strains of bacteria had the same effect on the viability of three different virulent nematodes.

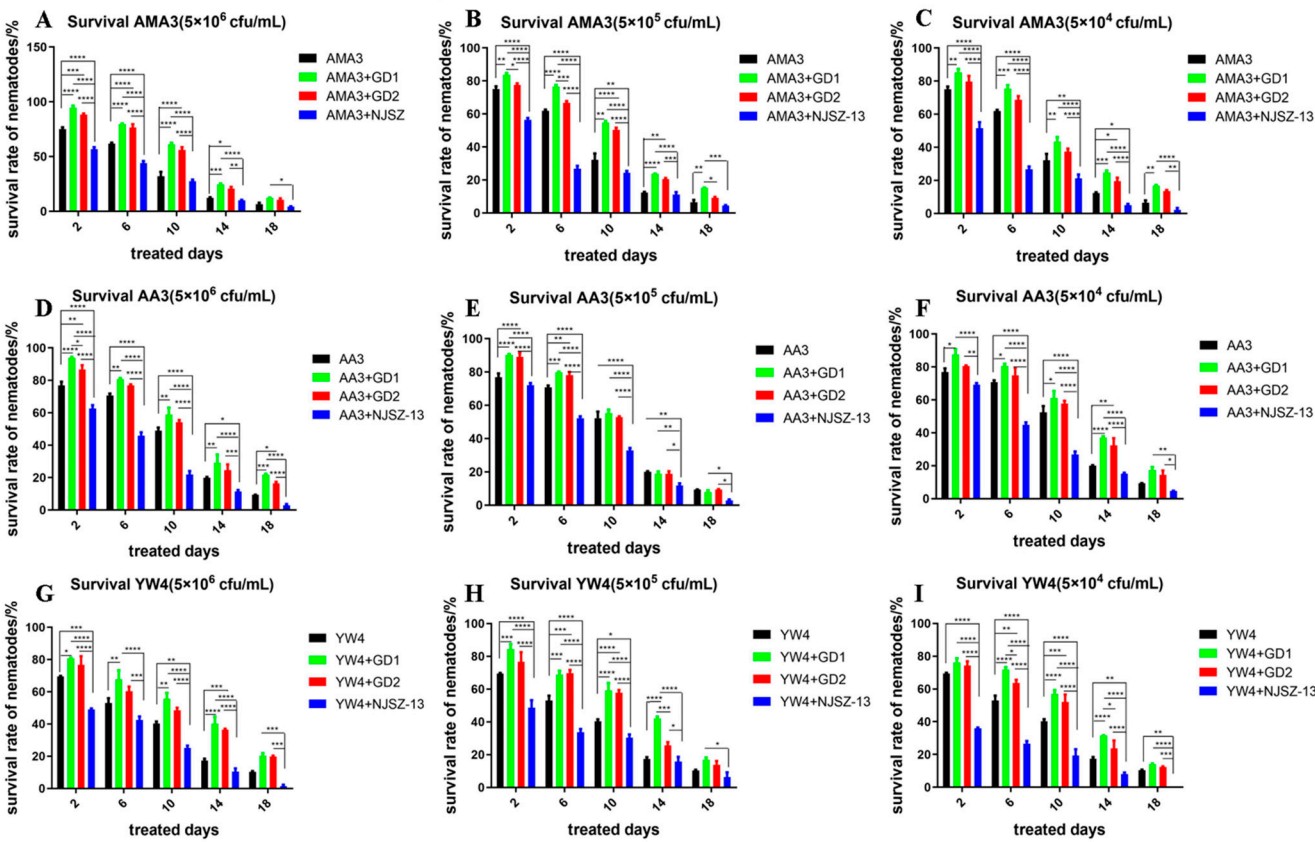

**Figure 1.** Effect trend of three different bacteria on the survival of different virulent pine wood nematodes (PWNs). (**A**–**C**) Effect trend of three strains of bacteria on AMA3 survival of PWNs. (**D**–**F**) Effect trend of three strains of bacteria on AA3 survival of PWNs. (**G**–**I**) Effect trend of three strains of bacteria on YW4 survival of PWNs. All data are presented as the means ± SEM. All of the results were analyzed by two-way analysis of variance: * $p < 0.05$, ** $p < 0.01$, *** $p < 0.001$, and **** $p < 0.0001$.

### 3.2. Fecundity of Different Virulent Nematodes Treated with Different Concentrations of Bacteria

The fecundity of high virulent AMA3, intermediate virulent AA3, and low virulent YW4 treated with $5 \times 10^6$, $5 \times 10^5$, and $5 \times 10^4$ CFU/mL of GD1 or GD2 was significantly different from that of the control groups (Figure 2), and different concentrations of GD1 and GD2 could promote the reproduction of PWNs. The fecundity of high virulent AMA3 and low virulent YW4 under $5 \times 10^6$, $5 \times 10^5$, and $5 \times 10^4$ CFU/mL of NJSZ-13 treatment was significantly lower than that of the control group (Figure 2), while there was no significant difference between the intermediate virulent AA3 under $5 \times 10^5$ and $5 \times 10^4$ CFU/mL of NJSZ-13 treatment and the control groups (Figure 2A,B). The fecundity of different virulent

nematodes treated with $5 \times 10^6$ CFU/mL of GD1 and GD2 was significantly higher in the GD1 treatment than that of the GD2 treatment (Figure 2C). The fecundity of low virulent YW4 under $5 \times 10^5$ CFU/mL of GD2 treatment was higher than that of the GD1 treatment (Figure 2B). GD1 and GD2 could significantly improve the fecundity of PWNs at three concentrations, while NJSZ-13 showed the opposite result. When the concentration of bacteria was $5 \times 10^6$ CFU/mL, the fecundity of low virulent YW4 treated with GD1 and GD2 was higher than that of $5 \times 10^5$ and $5 \times 10^4$ CFU/mL. The three strains of bacteria had the same effect on the fecundity of three different virulent nematodes.

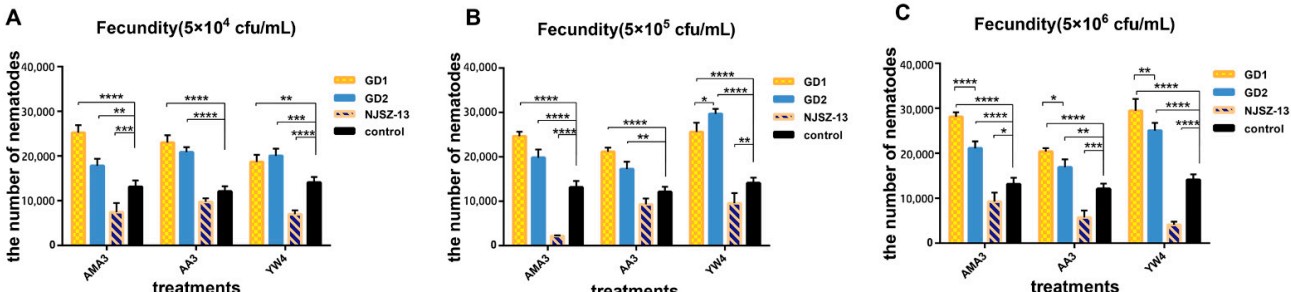

**Figure 2.** Effect of different concentrations of bacteria on the fecundity of pine wood nematodes (PWNs). (**A**) Fecundity of different PWNs treated with $5 \times 10^4$ CFU/mL bacteria. (**B**) Fecundity of different PWNs treated with $5 \times 10^5$ CFU/mL bacteria. (**C**) Fecundity of different PWNs treated with $5 \times 10^6$ CFU/mL bacteria. All data are presented as the means ± SEM. All of the results were analyzed by two-way analysis of variance: * $p < 0.05$, ** $p < 0.01$, *** $p < 0.001$, and **** $p < 0.0001$.

### 3.3. Mixed Inoculation of Bacteria and Nematodes to P. massoniana and L. kaempferi

The greenhouse inoculation experiment showed that the two-year-old pot seedlings of *P. massoniana* were infected with PWN in the AMA3 treatment group, AMA3 + GD1 treatment group, and AMA3 + GD2 treatment group, and the wilting symptoms were the same. The needles faded and yellowed at first, and then withered into reddish brown. However, the GD1 treatment group, GD2 treatment group, and the control group remained healthy, and the *P. massoniana* needles remained a healthy green (Figure 3). Forty days after inoculation, the disease incidence of *P. massoniana* inoculated with AMA3 was 40%, and the DI was 35; the disease incidence of AMA3 + GD1 was 40%, and the DI was 40; and the disease incidence of AMA3 + GD2 was 70%, and the DI was 70 (Table 3). Comparing the DI of *P. massoniana* under different treatments: AMA3 < a mixture of AMA3 and GD1 < a mixture of AMA3 and GD2.

**Table 3.** Disease incidence and disease index (DI) of *Pinus massoniana* inoculated with different treatments of pine wood nematodes. CK—control group.

| Days after Inoculation/d. | AMA3 | | AMA3 + GD1 | | AMA3 + GD2 | | GD1 | | GD2 | | CK | |
|---|---|---|---|---|---|---|---|---|---|---|---|---|
| | Disease Incidence/% | DI | Disease Incidence/% | DI | Disease Incidence/% | DI | Disease Incidence/% | DI | Disease Incidence/% | DI | Disease Incidence/% | DI |
| 5 | 0 | 0 | 0 | 0 | 0 | 0 | 0 | N/A | 0 | N/A | 0 | N/A |
| 10 | 0 | 0 | 0 | 0 | 0 | 0 | 0 | N/A | 0 | N/A | 0 | N/A |
| 15 | 10 | 2.5 | 10 | 2.5 | 20 | 10 | 0 | N/A | 0 | N/A | 0 | N/A |
| 20 | 10 | 7.5 | 10 | 10 | 30 | 22.5 | 0 | N/A | 0 | N/A | 0 | N/A |
| 25 | 20 | 20 | 40 | 35 | 40 | 35 | 0 | N/A | 0 | N/A | 0 | N/A |
| 30 | 20 | 20 | 40 | 35 | 50 | 37.5 | 0 | N/A | 0 | N/A | 0 | N/A |
| 35 | 20 | 20 | 40 | 40 | 70 | 67.5 | 0 | N/A | 0 | N/A | 0 | N/A |
| 40 | 40 | 35 | 40 | 40 | 70 | 70 | 0 | N/A | 0 | N/A | 0 | N/A |

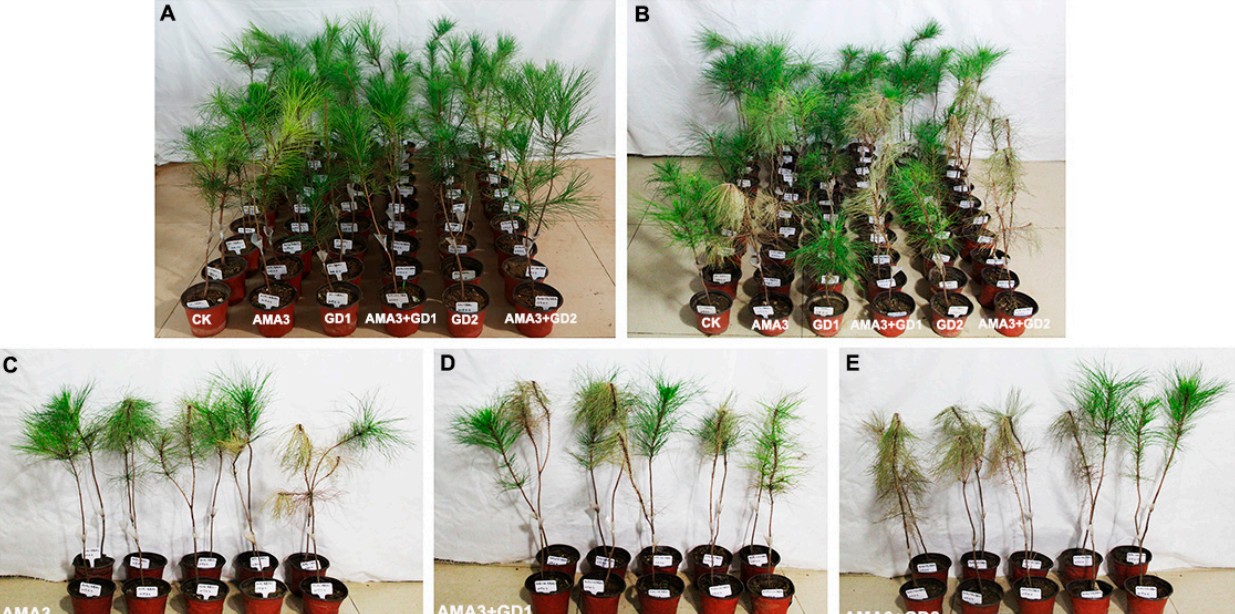

**Figure 3.** Symptoms of potted seedlings of *Pinus massoniana* (*P. massoniana*) after inoculation with *Bursaphelenchus xylophilus* (*B. xylophilus*), *B. xylophilus* treated with the bacterium from the body of the nematode (GD1), and *B. xylophilus* treated with the endophytic bacterium of *P. massoniana* (GD2). Each treatment of this experiment had ten saplings. (**A**) The symptoms of *P. massoniana* after inoculation for one day with AMA3, AMA3 treated with GD1, and AMA3 treated with GD2. (**B**) The symptoms of *P. massoniana* after inoculation for 40 days with AMA3, AMA3 treated with GD1, and AMA3 treated with GD2. (**C**–**E**) The symptoms of *P. massoniana* after inoculation for 40 days with AMA3 (**C**), AMA3 treated with GD1 (**D**), AMA3 treated with GD2 (**E**). Control group (CK) was only inoculated with 2 mL of sterile deionized water.

The needles faded and yellowed first, and then withered and turned reddish brown. However, the GD1 treatment group, GD2 treatment group, and control group all remained healthy, and the leaves of *L. kaempferi* remained a healthy green (Figure 4). Ten days after inoculation, the disease incidence of *L. kaempferi* inoculated with AMA3 was 40%, and the DI was 10; the disease incidence of AMA3 + GD1 was 40%, and the DI was 10; and the disease incidence of AMA3 + GD2 was 40%, the DI was 40. The GD2 + AMA3 treatment group was the first to show wilting symptoms. Fifteen days after inoculation, the disease incidence of *L. kaempferi* inoculated with AMA3 was 60%, and the DI was 15; the disease incidence of AMA3 + GD1 was 40%, and the DI was 35; and the disease incidence of AMA3 + GD2 was 60%, and the DI was 45. For the first time, there was a difference in the disease incidence between the treatment groups. Forty-five days after inoculation, the disease incidence of *L. kaempferi* inoculated with AMA3 was 60%, and the DI was 60; the disease incidence of AMA3 + GD1 was 60%, and the DI was 60; and the disease incidence of AMA3 + GD2 was 100%, and the DI was 85 (Table 4). Comparing the DI of *L. kaempferi* under different treatments: AMA3 < a mixture of AMA3 and GD1 < a mixture of AMA3 and GD2.

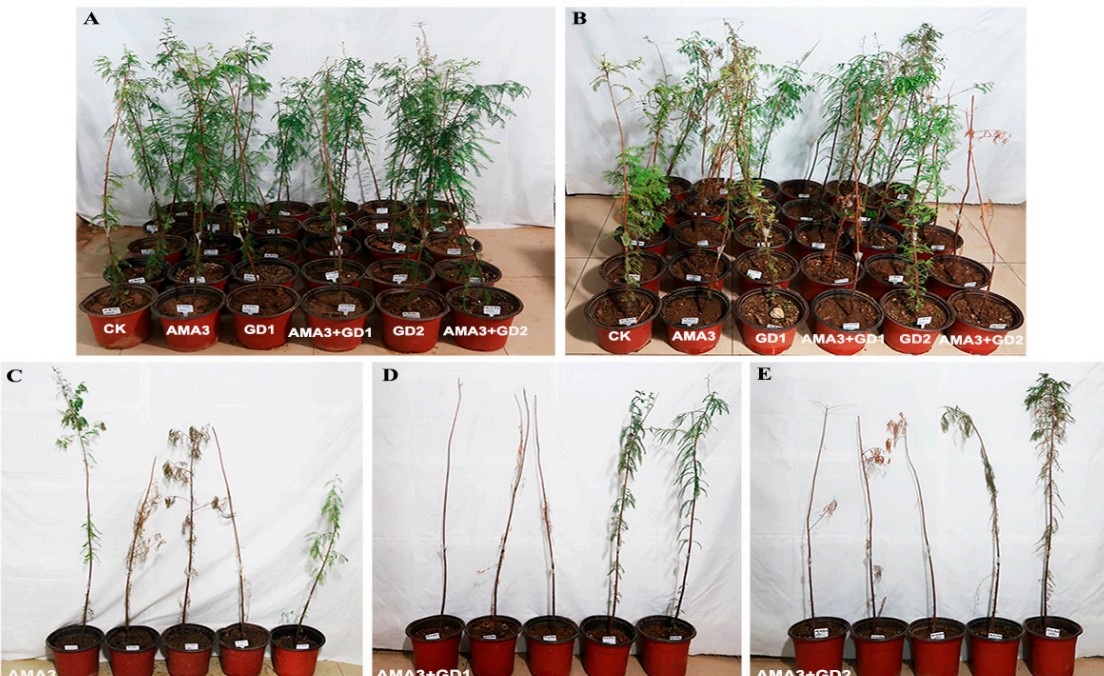

**Figure 4.** Symptoms of potted seedlings of *Larix kaempferi* (*L. kaempferi*) after inoculation with *Bursaphelenchus xylophilus*, *B. xylophilus* treated with the bacterium from the body of the nematode (GD1), and *B. xylophilus* treated with the endophytic bacterium of *P. massoniana* (GD2). Each treatment of this experiment had five saplings. (**A**) The symptoms of *L. kaempferi* after inoculation for one day with AMA3, AMA3 treated with GD1, and AMA3 treated with GD2. (**B**) The symptoms of *L. kaempferi* after inoculation for 45 days with AMA3, AMA3 treated with GD1, and AMA3 treated with GD2. (**C–E**) The symptoms of *L. kaempferi* after inoculation for 45 days with AMA3 (**C**), AMA3 treated with GD1 (**D**), AMA3 treated with GD2 (**E**). Control group (CK) was only inoculated with 2 mL of sterile deionized water.

**Table 4.** Disease incidence and disease index (DI) of *Larix kaempferi* inoculated with different treatments of pine wood nematodes. CK—control group.

| Days after Inoculation/d | AMA3 | | AMA3 + GD1 | | AMA3 + GD2 | | GD1 | | GD2 | | CK | |
|---|---|---|---|---|---|---|---|---|---|---|---|---|
| | Disease Incidence/% | DI | Disease Incidence/% | DI | Disease Incidence/% | DI | Disease Incidence/% | DI | Disease Incidence/% | DI | Disease Incidence/% | DI |
| 5 | 0 | 0 | 0 | 0 | 0 | 0 | 0 | N/A | 0 | N/A | 0 | N/A |
| 10 | 40 | 10 | 40 | 10 | 40 | 40 | 0 | N/A | 0 | N/A | 0 | N/A |
| 15 | 60 | 15 | 40 | 35 | 60 | 45 | 0 | N/A | 0 | N/A | 0 | N/A |
| 20 | 60 | 15 | 40 | 35 | 60 | 60 | 0 | N/A | 0 | N/A | 0 | N/A |
| 25 | 60 | 25 | 40 | 40 | 60 | 60 | 0 | N/A | 0 | N/A | 0 | N/A |
| 30 | 60 | 45 | 40 | 45 | 80 | 65 | 0 | N/A | 0 | N/A | 0 | N/A |
| 35 | 60 | 50 | 60 | 60 | 80 | 70 | 0 | N/A | 0 | N/A | 0 | N/A |
| 40 | 60 | 60 | 60 | 60 | 80 | 70 | 0 | N/A | 0 | N/A | 0 | N/A |
| 45 | 60 | 60 | 60 | 60 | 100 | 85 | 0 | N/A | 0 | N/A | 0 | N/A |

### 3.4. Differential Gene Expression Analysis

Satisfy the screening conditions of differential genes, $p < 0.05$ and $|log2FC| > 1$. which are considered to be differentially expressed genes. According to this principle, this study screened a total of 262 differential genes, of which 121 were up-regulated genes and 141 were down-regulated genes. In the comparison between the CK group and the T1 (the transcriptome of AMA3 processed by GD1) group, there were 43 up-regulated genes and 79 down-regulated genes (Figure 5A). In the comparison between the CK group and the T2 (the transcriptome of AMA3 processed by GD2) group, there were 49 up-regulated genes and 53 down-regulated genes (Figure 5B). Further research found that the top six genes with the most significant differential expression were *Hsp72*, *Hsp70*, *sHsp 21*, the *DSBA*

oxidoreductase domain protein, the organic solute transport α-1 protein (osta-1), and the hypothetical protein. These genes were associated with longevity-regulating pathway worm, MAPK signaling pathway, drug-metabolism cytochrome P450 and Bile secretion, respectively (Table 5).

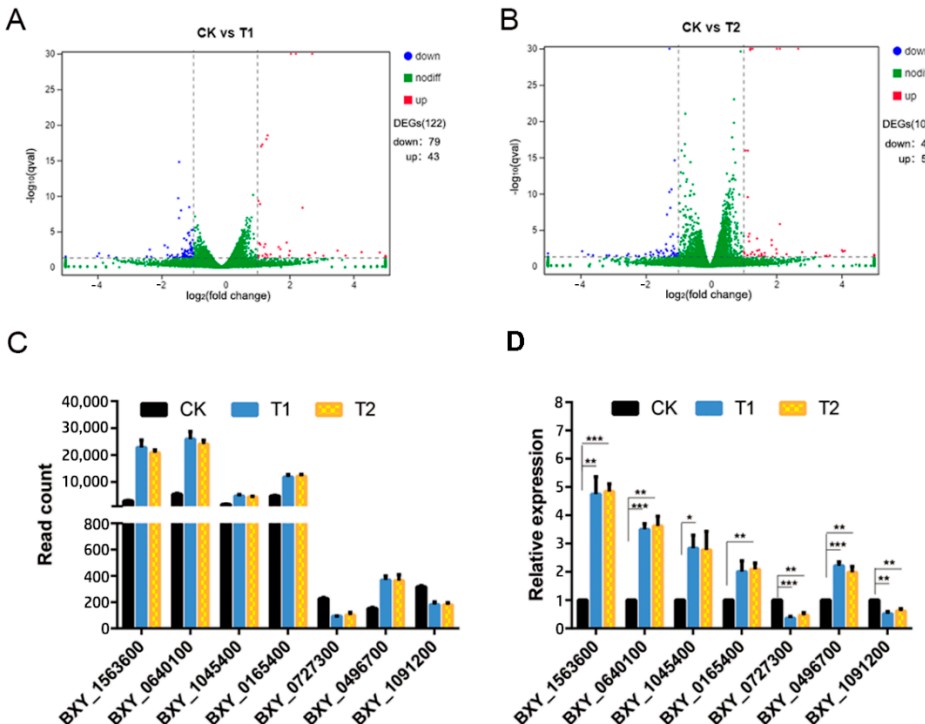

**Figure 5.** Differential gene expression analysis. CK: the transcriptome of AMA3; T1: the transcriptome of AMA3 processed by GD1; and T2: the transcriptome of AMA3 processed by GD2. (**A**) The comparison of CK and T1. (**B**) The comparison of CK and T2. The blue circles represent down-regulated genes, and the red squares represent up-regulated genes, respectively. (**C**) The result of top 6 genes expression count in RNA-seq. (**D**) The result of relative expression of q-PCR. * $p < 0.05$, ** $p < 0.01$ and *** $p < 0.001$.

**Table 5.** Relative pathways of 6 differentially expressed genes.

| ID | Gene Name | KEGG-ko | Pathway |
|---|---|---|---|
| BXY_1563600 | *Hsp 72* | ko04010 | MAPK signaling |
| BXY_0640100 | *Hsp 70* | ko04010 | MAPK signaling |
| BXY_0165400 | *sHsp 21* | ko04212 | Longevity-regulating pathway worm |
| BXY_0727300 | *DI09 29p80* | ko00982 | Drug-metabolism cytochrome P450 |
| BXY_0496700 | *osta-1* | ko04976 | Bile secretion |
| BXY_1091200 | *CBG 01395* | ko04976 | Bile secretion |

To further verify the accuracy of the RNA sequencing results, the top six genes were performed using the q-PCR. The result showed that the relative expressions of the top six genes were basically consistent with the trend of transcriptome sequencing (Figure 5C,D).

## 4. Discussion

Studies have shown that the remarkable diversity of PWN-carried bacteria and endo-phytes in pine trees [32,33], and the functional relationship between PWNs and bacteria has been proposed [34], but the bacterial hypothesis still seems to involve many issues that need to be clarified [35]. There is a subtle relationship between bacteria and PWNs,

likely contributing to survival, fecundity, and host adaptability. This study aimed to further clarify the role of bacteria in PWD. Zhao and Lin [36] showed that there was a symbiotic relationship between PWNs and *Pseudomonas*, which could significantly improve its fecundity. Tan and Feng [27] showed that strains GD1 and GD2 could also significantly improve the survival and fecundity of PWN. Furthermore, the research shows that the bacteria *Serratia* can help PWNs survive under long-term oxidative stress conditions [37]. This study revealed that PWN-associated bacteria and pine endophytic bacteria can significantly improve the survival and fecundity of PWNs with different virulence, and the effect of PWN-associated bacteria is more significant than that of pine endophytic bacteria. In addition, bacteria with nematicidal activity significantly inhibit the activity and fecundity of PWNs with different virulences. Further study showed that there were significant differences in the effects of different bacteria on the survival and fecundity of PWNs at different concentrations. Interestingly, the effect did not change irrespective of PWN virulence, suggesting a universality of bacterial effects on PWNs, paving the way for a new theoretical basis for the prevention and control of PWD.

To verify whether there is a difference in the effects of PWN-associated bacteria and *P. massoniana* endophytic bacteria on the host adaptability of PWNs between pine and non-pine genus, we compared the DI of PWN-associated bacteria and endophytic bacteria in *P. massoniana* and *L. kaempferi*. When the bacteria were inoculated separately on the host, the host did not show symptoms of PWD, but the PWD development speed of mixed inoculation (of bacteria) with PWNs is faster than that of PWNs alone. Similarly, Chi et al. [38] found that only mixed inoculation of sterile PWNs and toxic *Pseudomonas fluorescens* (*P. fluorescens*) could cause symptoms of PWD in aseptic *P. thunbergii* seedlings, while sterile PWNs and *P. fluorescens* alone could not cause wilting of *P. thunbergii*. Furthermore, Vicente et al. [17] found that four strains of bacteria associated with PWNs were inoculated separately into *P. pinaster*, which also showed wilting symptoms of PWD. The results showed that different bacteria had different effects on the host adaptability of PWNs. Among them, the host adaptability of pine endophytic bacteria to nematodes may be more significant than that of nematode-associated bacteria. In previous studies, we found that DI was significantly reduced when mixing inoculation with pine wood nematodes and NJSZ-13 [26]. There were differences in the effects of PWN-associated bacteria and *P. massoniana* endophytic bacteria on the host adaptability of PWNs in pine and non-pine genera. In addition, after the first inoculation, *L. kaempferi* did not show obviously susceptible symptoms, so the second reinoculation was carried out in this study, and *L. kaempferi* showed wilting symptoms. As we all know, PWD is often accompanied by the repeated transmission of vector insects in the natural environment. In the inoculation of *L. kaempferi*, perhaps because of its high resistance, the symptoms of the first inoculation are not obvious. However, by repeated inoculation in the control and the treatment, we can still observe similar results as in the *P. massoniana* inoculation experiment, which indicated that strains GD1 and GD2 may improve the adaptability of PWNs to non-pine hosts. Can we boldly assume that bacteria play the role of conditional pathogens [39] in PWD? In the healthy host microbial community, bacteria are harmless or even beneficial to the host; however, once the PWN invades the host, bacteria may promote the PWD by expressing their own detoxification genes or improving the adaptability of PWN to the host [19,20,37]. In the inoculation test, the effect of GD2 on the PWD was more obvious than that of GD1, indicating that pine endophytic bacteria may play a role in nematode-adapting plant hosts.

Our result showed that the AMA3 population treated with GD1 and GD2 had a great impact on the gene expression of *B. xylophilus* by RNA sequencing. Genes such as heat shock protein 72 (*Hsp72*) and small heat shock protein 21 (*sHsp21*) were significantly up-regulated. Further study found that some genes are associated with some important pathways in PWNs. *Hsps* are the downstream genes of the *Hsf-1* gene in the longevity regulation pathway, which can inhibit protein aggregation, thereby prolonging the lifespan of the body [40]. This may explain the increase in the survival rate of PWNs treated with GD1 and GD2. The *Hsp72* protein in the MAPK signaling pathway can inhibit the

expression of inflammatory factors in the JNK pathway, thereby reducing the occurrence of inflammation. Previous research showed that the *Hsp72* protein cooperates with *Hif-1* in *Caenorhabditis elegans* to reduce heat stress damage [41]. *Hsps* can promote protein synthesis and transportation in the body, and enhance the body's tolerance to adversity stress and adaptability to different living environments. It indicated that the PWNs AMA3 treated with GD1 and GD2 increased the expression of *Hsps*, and improve the body's adaptability to cope with the external environment, thereby increasing its own viability. We speculate that the reason why nematodes can coexist with bacteria and aggravate the PWD may be related to the up-regulated expression of some genes involved in the regulation of the longevity-regulating pathway worm, MAPK signaling pathway, and drug-metabolism cytochrome P450 after treatment with strains GD1 and GD2.

In conclusion, the effects of different bacteria on the survival and fecundity of PWNs were different at different concentrations, and the effect did not change with PWNs of different virulences. The effects of different bacteria on the host adaptabity of PWNs were different. Among them, the host adaptability of pine endophytic bacteria to nematodes may be more significant than that of nematode-associated bacteria, and the effect does not change with different hosts. The role of bacteria may not only be related to the existence of nematodes, but may also depend on the relationship between bacteria and hosts. These basic results support the possibility that bacteria play a role in the host adaptability of PWNs. Our findings provided new information about the interaction between PWNs and bacteria. RNAi can then be performed on some related genes that may be responsible for improving adaptive functions. In this way, the role of bacteria in the development of PWD is explored.

**Author Contributions:** Y.Y. and Z.Y. have contributed equally to this work and share first authorship. J.T. and J.Y. conceived and designed the experiments; Y.Y., Z.Y. and Y.C. carried out the experiments and data analysis; and Z.Y. and Y.Y. wrote the paper with the help of all authors. All authors have read and agreed to the published version of the manuscript.

**Funding:** This research was supported by the National Key Research and Development Program of China (2021YFC1400900 and 2018YFC1200400).

**Institutional Review Board Statement:** Not applicable.

**Informed Consent Statement:** Not applicable.

**Data Availability Statement:** The datasets generated and analysed during the current study are available in the Sequence Read Archive (SRA) repository, [https://www.ncbi.nlm.nih.gov/bioproject/PRJNA760241], (accessed on 3 September 2021).

**Acknowledgments:** We are grateful for the assistance of all staff and students at the Institute of Forest Protection, Nanjing Forestry University, Nanjing, China. We thank Accdon for its linguistic assistance during the preparation of this manuscript.

**Conflicts of Interest:** The authors declare that they have no competing interest.

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
