# Peer review of "Effects of Bacillus cereus on Survival, Fecundity, and Host Adaptability of Pine Wood Nematode"

_diversity, doi:10.3390/d15040566_

Round 1
Reviewer 1 Report
Original and relevant research exploring the effect of three strains of Bacillus cereus, some of which had a positive effect on the survival, fecundity and adaptability the pinewood nematode to its host. This study sheds new light onto the pathogenic mechanisms of the nematode and how bacteria can modulate its pathogenicity and survival, paving the way for new management strategies of pine wilt disease. I provide some suggestions to improve the manuscript.
Major comments:
Abstract & Introduction: well-structured, objective approach to the state of the art and appropriate references.
Material & Methods: experimental design is good and methods are adequate. Proper references to previously published methodology. Good statistical analysis. However, I picked up some incongruences in the figures that I point out in the comments below.
Results & Discussion: results, including tables and figures, are very nicely presented. However, I picked up some incongruences in the figures that I point out in the comments below. Logical interpretation of results and conclusions.
Specific comments:
Abstract
Line 15: remove the second “strains” word
Line 17: at different concentrations
Introduction
Line 30: The pine wood nematode (PWN), Bursaphelenchus xylophilus, [...] – remove “(B. xylophilus)”
Line 38: species of the Pinus genus
Lines 40-41: I would rephrase to “In December 2018, PWD was reported on non-Pinus host Larix gmelinii in China.” Remove “(L. gmelinii)”.
Line 42: when writing the name of a species at the beginning of a sentence, write it out: Larix gmelinii
Lines 67-68: “It is worth mentioning that the role played by bacteria 67 in this process.” – this sentence is out of place, I think
Line 73: PWN endophytic bacteria – endophytic means inside the plant; in this case, you should use PWN (endo)symbiotic bacteria, or PWN-associated bacteria (in case they are not symbiotic)
Material and Methods
Section 2.2.: How were the bacteria maintained? Which medium was used?
Lines 107-109: This sentence is confusing: “Fresh single colony GD1, GD2, 107 and NJSZ-13 were selected and put into 150 mL flasks containing 50 mL nutrient broth medium, shaken at 200 r/min at 27°C for 12h put into aseptic 50 mL centrifuge tubes. And centrifuged at 6500 r/min for 3 min.” – single colonies were obtained from agar plates (see my previous comment) and then put into 150 mL flasks with 50 mL nutrient broth overnight. On the next day, the bacterial suspension was transferred to a sterile 50 mL centrifuge tube and centrifuged at 6500 r/min for 3 min. Is that it?
Line 112: “plate colony count” – with a hemocytometer?
Line 113: I would rephrase the title to “Viability of nematodes with contrasting virulence under different bacterial treatments”, to avoid repetitions
Line 114: [...] 2000 PWNs of AMA3 were mixed [...]
Line 130: “The mixture of 0.2 mL bacteria” – isn’t it 2 mL?
Lines 180-181: This information is repeated from the previous section (2.6): “Total RNA of different groups was extracted using Trizol reagent kit (Invitrogen, 180 Carlsbad, CA, USA) according to the manufacturer’s protocol.” You can remove it entirely.
Line 205: All data were expressed as means ± standard error of means (SEM)
Results
Lines 217-219: “Under the treatment of 5×10^4 CFU/mL strains, the result of the control group, AMA3+GD1, and 218 AMA3+GD2 treatment groups decreased most rapidly between day 6 and day 10 (Fig. 1A)” – Fig 1A shows the treatment 5×10^6 CFU/mL and not 5×10^4 CFU/mL, as referred in the text
Lines 220-224: Same as my previous comment; the text is not in line with what is shown in the referred figures. Please, fix accordingly.
Line 229: “All data are presented as the means”. Some info is missing: means ± SEM. All of the results were analyzed by two-way ANOVA, *p<0.05, **p<0.01, ***p<0.001, ****p<0.0001 (just like you did in lines 274-275).
Figure 1: what happened to the nematodes in the YW4+NJSZ-13 group (5×10^4 CFU/mL) at day 18? They were all dead?
Lines 243-245: Fig. 1G shows the treatment 5×10^6 CFU/mL and not 5×10^4 CFU/mL, as referred in the text
Lines 248-249: The treatments 5×10^5 and 5×10^6 CFU/mL strains are shown in Fig. 1G-H. Correct this, please
Line 273: “5×105” – the second 5 needs to be uppercased
Line 279: infected with PWN and not PWD; you may say that seedlings had symptoms of PWD, if that was the case. Either one (infected by PWN) or the other (showed symptoms of PWD).
Line 286: Table 2 is mentioned in the text, but it really refers to Table 3
Note: Fig. 3 appears first in the text, so I think it would be better to present it above Table 3 (like you did for Fig.4 and Table 4)
Line 334: you need to mention what T1 stands for the first time you refer it in the text
Figure 5: the letter D is missing for graph on the lower right (and the description needs to be added to the figure legend as well)
Discussion
Lines 372-374: I would rephrase the sentence to: Interestingly, the effect did not change irrespective of PWN virulence, suggesting a universality of bacterial effects on PWNs, paving the way for a new theoretical basis for the prevention and control of PWD.
Lines 380-381: “[...] but the PWD development speed of mixed inoculation (of bacteria) with PWNs is faster than that of PWNs alone.”
Line 409: “Our result showed that the AMA3 strains” – change strains to population or something similar
Reviewer 2 Report
Effects of Several Strains of Bacteria on Survival, Fecundity, and Host Adaptability of Pine Wood Nematode with Different Virulence
The study “Effects of Several Strains of Bacteria on Survival, Fecundity, and Host Adaptability of Pine Wood Nematode with Different Virulence” sounds very interesting. I have a few recommendations to further improve the quality of the manuscript and request the authors to consider them.
1. I would suggest a thorough revision of the manuscript (including the abstract) by a person with English language fluency
2. Title: The title is a bit vague and does not reflect the results. Though technically it explains what has been done, I suggest the authors to reconsider the title of the manuscript
3. Line number 50 if there is a name for the wilt toxin or specific nature, it is better to use that as the current line is confusing.
4. Were the three strains GD1, GD2 and NJSZ-13 screened for production of wilt toxin?
5. Line 121 is it ‘observed’? rather than ‘absorbed’
6. In figure 1, the strain NJSZ-13 is found to be highly nematicidal (at 5*104 cfu/mL) against less virulent pine wood nematode YW4. I understand the range of virulence in the PWNs were in infecting the plants could there be any relationship with susceptibility towards nematicidal compounds/antagonistic microorganisms?
7. from figure 2, it is hard to understand any relationship between the microbial population strength and fecundity except for the antagonistic strain NJSZ-13. Would there be any particular reason for its high antagonisms at 5*105 cfu than the other studies concentrations? Seems like an interesting phenomenon
8. Line 373. Do the author mean PWNs with variable virulence by ‘different PWNs’?
9. Line 379 would there be any reason for this? Since the wilt toxin is believed to be produced by the bacterium, is there any known mechanism of triggering by the PWN for bacterial synthesis of the wilt causing toxin?
10. Line 389-390 Please explain what ‘mixed’ means here. It is possible to go the reference. However, since it might be helpful for this discussion please elaborate on ‘mixed’.
11. Lines 405-406 also present an interesting phenomenon as we cannot ascertain what kind of microorganism would be responsible for PWD as GD2 is a plant endophytic bacteria.
12. Line 413. Consistency in sHsps
13. Line 418 please change ‘some research’ to ‘A previous research’ since only one reference has been provided
14. Generally, qPCR validation is not compulsorily required for RNA-Seq studies. Please check out the publication below 10.1016/j.bioflm.2021.100043
15. Line 430 Do the author mean PWNs with variable virulence by ‘different PWNs’?
Reviewer 3 Report
The article submitted for review concerns the effect of selected bacterial strains on Bursaphelenchus xylophilus isolates. Research on this nematode species is particularly important. B. xylophilus is one of the most dangerous pests of forest ecosystems around the world. A good understanding of the biology and biological interactions of this nematode species is crucial for reducing environmental and economic losses.
The article was rather well prepared. The results obtained support the authors' conclusions. Before being accepted for publication, the manuscript should be carefully read and rewritten, as some passages are not very comprehensible.
More important comments:
L13: why further? Even if the authors have studied the problem before, they should not refer to it at the beginning.
L30: remove (B. xylophilus), this repetition is not necessary.
L42: remove (L. gmelinii), this repetition is not necessary.
L46: “The pathogenic mechanism of PWNs has always been controversial” – explain.
L65-66: change “adaptability” to “tolerance”.
L68: It is worth mentioning that the role…” – check grammar or out of context.
L74: as above
L87: Briefly explain what the differences in nematode virulence were.
L95: Briefly explain how the bacteria were identified.
L103: “The liquid” – rather “nematodes suspension”
L105-112: explain the purpose of the treatments, check the grammar
L132: why gradient incubator? Explain.
L135: The authors should clearly indicate the size of the study groups, their number (N), and the number of replicates.
L156: symptoms of what?
L225: enlarge the charts, in their current form they are not readable.
L230- authors should refer to statistical analyses, indicating significant differences, p level.
L346: no marking of D chart, no description.
Round 2
Reviewer 2 Report
Thanks for addressing my concerns. I see that the manuscript can be more interesting to read if the language could be improved and suggest the authors to have the final form revised by an English language expert. It would significantly improve the reading experience.